# Tularemia Goes West: Epidemiology of an Emerging Infection in Austria

**DOI:** 10.3390/microorganisms8101597

**Published:** 2020-10-16

**Authors:** Stefanie Seiwald, Anja Simeon, Erwin Hofer, Günter Weiss, Rosa Bellmann-Weiler

**Affiliations:** 1Department of Internal Medicine II, Infectious Diseases, Immunology, Rheumatology, Pneumology, Medical University of Innsbruck, 6020 Innsbruck, Austria; stefanie.seiwald@tirol-kliniken.at (S.S.); anja.simeon@i-med.ac.at (A.S.); guenter.weiss@i-med.ac.at (G.W.); 2Institute for Veterinary Disease Control, Austrian Agency for Health and Food Safety (AGES), 2340 Mödling, Austria; erwin.hofer@ages.at

**Keywords:** *F. tularensis*, tularemia, Austria, epidemiology

## Abstract

The zoonotic disease tularemia is caused by the Gram-negative bacterium *Francisella tularensis*, with the two major subspecies *tularensis* and *holarctica* being responsible for infections in humans and animals. The *F. tularensis* subspecies *holarctica* is less virulent and prevalent in Europe and Asia. Over the last few centuries, few epidemic outbreaks and low numbers of infections have been registered in the eastern part of Austria, specifically in the provinces of Lower Austria, Burgenland, and Styria. The reported infections were mostly associated with hunting hares and the skinning of carcasses. Within the last decade, ticks have been identified as important vectors in Tyrol and served as first evidence for the spread of *F. tularensis* to Western Austria. In 2018, the pathogen was detected in hares in the provinces of Tyrol, Vorarlberg, and Salzburg. We presume that *F. tularensis* is now established in most regions of Austria, and that the investigation of potential host and vector animals should be spotlighted by public institutions. Tularemia in humans presents with various clinical manifestations. As glandular, ulceroglandular, and typhoidal forms occur in Austria, this infectious disease should be considered as a differential diagnosis of unknown fever.

## 1. Introduction

Tularemia is a zoonotic disease caused by *Francisella tularensis* (*F. tularensis*), a Gram-negative, facultative intracellular bacterium, which was first isolated from ground squirrels in Tulare County, (California, United States) by McCoy and Chapin in 1911 [1,2]. The first bacteriologically confirmed case was described by Wherry and Lamb in Ohio in 1914 [3].

The pathogen *F. tularensis* is currently divided into four subspecies: *Francisella tularensis* subsp. *tularensis*, *Francisella tularensis* subsp. *holarctica*, *Francisella tularensis* subsp. *mediasiatica*, and *Francisella tularensis* subsp. *novicida* [4]. Initially, *F. tularensis* subsp. *novicida* was classified as a distinct species, *F. novicida*, because of phenotypic differences and less fastidious growth requirements compared to *F. tularensis* [5]. In the 1980s, based on DNA–DNA hybridization experiments, it was then suggested to reassign *F. novicida* as a subspecies to *F. tularensis* because of a high degree of genetic relatedness [6]. To date, there has not been a clear decision on the correct nomenclature and both names are still in use [7].

The subspecies *tularensis* (type A) and *holarctica* (type B, former subsp. *palaearctica*) are the main causative agents for tularemia in humans and animals. The less virulent type B occurs throughout the Northern hemisphere, with predominance in Europe and Asia, and is further divided into biovar I (erythromycin sensitive, prevalent in Western Europe), biovar II (erythromycin resistant, prevalent in Northern and Eastern Europe), and biovar *japonica* (prevalent in Japan, China, and Turkey) [8,9,10,11]. In contrast to *F. tularensis* subsp. *holarctica* biovar I and II, biovar *japonica*, like *F. tularensis* subsp. *tularensis*, produces acid, not only in glucose-containing media, but also in glycerol-containing media. By amplifying variable-number tandem repeats (VNTRs), *F. tularensis* subsp. *holarctica* was first separated into five major clades (B.I, B.II, B.III, B.IV, and B.V) [12]. Later on, assays based on a canonical single nucleotide polymorphism (canSNP) and canonical insertions/deletions (INDELs), which circumvented the previous classification’s disadvantage for phylogenetic studies, led to a reduction to the four major clades: B.12 (B.I), corresponding to biovar II Eastern and Central European strains; B.4 (B.II), corresponding to North American strains; B.6 (B.IV), corresponding to biovar I Western European strains; and B.16 (B.V), corresponding to strains belonging to biovar *japonica* [10,13,14,15,16,17,18]. Recent reports of tularemia cases in humans and ring possums in Australia caused by *F. tularensis* subsp. *holarctica* have confirmed the presence of the pathogen also in the Southern hemisphere [19,20].

The highly virulent type A appears to be restricted to North America, and causes almost 70% of human cases on this continent [1,9,21]. Although some strains of *F. tularensis* subsp. *tularensis* were isolated from mice and fleas in the Danube region between 1978 and 1996, human tularemia infections in Europe have only been caused by *F. tularensis* subsp. *holarctica* so far [22,23]. The obtained type A isolates were later assigned to a laboratory strain by genome sequencing, and it was assumed that anthropogenic activities, such as the disposal of laboratory waste or the escape of infected animals, had led to an environmental contamination [24].

Infections with *F. tularensis* have been reported in a wide range of vertebrates, amphibians, fish, and invertebrates [25,26]. Furthermore, isolation of *F. tularensis* DNA in water and sediment indicates silent persistence in the environment, likely originating from dead affected animals or their excrements [23,27]. It has been experimentally proven that contaminated silt may remain infectious for up to two months [28]. According to the highly complex ecology of *F. tularensis*, transmission to humans may occur via different routes, such as direct contact with infected animals (e.g., during animal processing, through ingestion of uncooked meat, and animal bites), arthropod-borne (ticks, horseflies, and mosquitos) and through the consumption of contaminated water or inhalation of contaminated soil, as it may occur during farming works [23,29,30,31,32]. 

The European hare (*Lepus europaeus*), also known as the brown or field hare, is considered an important host of *F. tularensis* and a common vector for the pathogen’s transmission to humans in Europe [26,27,33]. In this species, variable clinical courses ranging from acute deadly septicema to protracted courses with only subacute lesions in various organs are reported. In case of chronic infection, European hares may serve as long-term reservoirs for *F. tularensis*, thus bearing a persistent risk of transmission to humans, either directly or via vectors [34,35,36]. Histopathological examinations demonstrated differences in the pathogenicity of clade B.FTNF002-00 (subgroup of B.6 and specific for Western Europe) and clade B.13 (subgroup of B.12 and specific for Central and Eastern Europe) in European brown hares [18,37]. While infections with strains of clade B.13 were reported to be associated with polyserositis, affecting the kidneys, pleura, and pericardium, histopathological findings in hares infected with B.FTNF002-00 have been almost invariably characterized by splenitis and hepatitis [34,37]. These results are in accordance with further observations in experimentally infected rats, showing significant differences in weight loss, mortality rate, and time to recovery between the two genotypes [38].

Rodents are very susceptible to *F. tularensis*, and commonly present with severe infection, leading to early death [39]. Nevertheless, experimental studies showed that infected voles may also show a protracted course of disease with chronic nephritis and bacteriuria, and could therefore also serve as a prolonged source of environmental contamination [40]. In the case of an epizootic event, water, soil, and foodstuffs may be contaminated through carcasses, faeces or urine from infected individuals [28,41]. Several large outbreaks of human tularemia have been reported due to ingestion of contaminated water, some of them supposedly associated with infected rodents, while others showed a demonstrable link [42,43,44,45]. Moreover, in Spain, a large pneumonic outbreak was associated with farming activities due to the inhalation of contaminated aerosols, as well as direct contact to common voles (*Microtus arvalis*) [46,47].

Moreover, ticks appear to play a key role in the ecology of tularemia among arthropods, as they may carry pathogens over several years and life stages, thus maintaining enzootic tularemia foci between epizootic periods [27,48,49,50]. The genus *Dermacentor reticulatus* seems to be the most frequent carrier of *F. tularensis* in Central Europe [51,52,53]. 

In Sweden, mosquito-borne infections, which are related to the aquatic life cycle, are a common route for transmission of tularemia in humans and can lead to major outbreaks [54,55,56]. Studies have also provided evidence that mosquito larvae may be infected with *F. tularensis* via water, possibly by ingesting predatory protozoa [57].

Defined roles for the different potential host species in the ecology of tularemia are not yet sufficiently clarified, and may depend on geographical factors, as well as on the susceptibility and sensitivity of respective organisms to the pathogen [58]. Although recent observations provide evidence for differences in virulence among the specific lineages of *F. tularensis* subsp. *holarctica*, there is no clearly defined preference of certain lineages for particular host species [59]. It is assumed that most of the animals do not serve as amplifying hosts, increasing the basic reproduction number of *F. tularensis*, but are “incidental” dead-end hosts, and therefore may not play a role in the persistence and spread of the pathogen [58].

Transmission of tularemia from domestic animals to humans has not yet been documented in Europe, but reports of cat- and dog-related tularemia in humans exist in the United States. There, in a nationwide case study on human tularemia between 2006 and 2016, about 3% of infections were classified as canine-transmitted [60,61]. Furthermore, in a recent study, almost 50% of human tularemia cases in the United States were reported to be cat-associated [62]. Occasional outbreaks were observed in sheep during the lambing season, due to their feeble physical condition [49]. To date, there is no evidence for human-to-human transmission in Europe. However, due to its high virulence and its feasibility to be spread via aerosols, *F. tularensis* is classified as a category A potential agent for bioterrorism [23,63].

In humans, following a short incubation time of 3 to 5 days (maximum 2 to 3 weeks), tularemia may present with unspecific, flu-like symptoms, including fever, lymphadenopathy, headache, chills, myalgia, and arthralgia, and is therefore often misdiagnosed, especially in areas with low incidence [63,64]. The clinical presentation depends on the location of bacterial entry into the body. The six major clinical manifestations are comprised of ulceroglandular, glandular, oropharyngeal, oculoglandular, pneumonic, and typhoid forms. Ulceroglandular and glandular tularemia are acquired by direct contact with infected animals or are vector-borne, and are the most frequent clinical manifestations in European countries, covering more than 95% of human cases [23,63]. The pneumonic form is mostly related to farming activities, and is caused by the inhalation of aerosols, originating from carcasses of rodents and lagomorphs or infected dead animals [65].

The diagnosis of tularemia is confirmed via serological, PCR, and antigen testing, as well as from culturing [63]. Concerning therapy, aminoglycosides, tetracyclines, and fluorochinolones are the antimicrobial drugs of choice, due to well-established clinical efficacy and the lack of reports on resistance [66,67,68]. For patients with severe infection, aminoglycosides are the recommended therapy because of high cure and minimal relapse rates [69]. Due to the potential of *F. tularensis* as a bioterrorism agent, in recent decades increased research has been conducted to develop an efficient and safe vaccine. Among previous approaches, the live attenuated vaccine “Live Vaccine Strain” (LVS) was particularly promising, but has shown insufficient protection against strains of type A and has not been licensed in the United States or Europe, due to safety concerns regarding toxicity. Currently, promising research focusses on new genetically modified, live-attenuated vaccine strains and subunit vaccines with improved safety [70,71,72].

Tularemia is a notifiable disease in most European countries, meaning that both confirmed infection and death, as well as suspected cases, must be reported to the respective authorities. While Cyprus, Greece, Iceland, Ireland, Luxembourg, Malta, and the United Kingdom (only imported cases) are considered to be free of tularemia, the disease occurs in the other European countries with varying incidence [73,74]. Sweden and Norway recorded 40% in 2017 and 45% in 2018 of all reported cases of human tularemia across the member states of the European Union (EU) and the European Economic Area (EAA) [74,75].

## 2. Aims and Methods

The aim of this review is to focus on the epidemiology of tularemia in Austria using available data from the past and present. In addition to a literature search in the databases PubMed, VetMed, and UpToDate with a selection of reviews, case reports, prevalence studies, and guidelines, the annual statistics on human tularemia cases from 1990–2019, available from the Federal Ministry of Labour, Social Affairs, Health and Consumer Protection were included. The research also covered the website of the Austrian Agency for Health and Food Safety GmbH (AGES). Informed consent for publication of the cases and images was obtained from the Tyrolean patients.

## 3. Results

In Austria, direct or indirect contact with infected animals and tick bites represent the main transmission modes of tularemia to humans. In 2014, a nationwide cross-sectional serological analysis of 526 healthy adults from Austria found that 0.5% had detectable antibodies against *F. tularensis* [76]. Tularemia sporadically occurs in all counties of Austria, and until 2015 most cases were reported from the provinces of Lower Austria, Burgenland, Vienna, Styria, and Upper Austria, consequently regarded as endemic areas [23]. According to the predominant modes of transmission, the most frequent clinical manifestations of human tularemia in Austria include the ulceroglandular and the glandular forms. However, one case of a typhoid form with secondary pulmonary tularemia was recently reported in a forester from Lower Austria. The suspected route of infection in this case was the inhalation of contaminated dust during forest work [77]. Strains of *F. tularensis* detected in Austria are commonly assigned to the phylogenetic group B.13 (subclade of major clade B.12), corresponding to biovar II [10]. However, recently a strain belonging to biovar I was isolated from a person suffering from tularemia in the province of Tyrol, Western Austria [78].

### 3.1. Human Cases

#### 3.1.1. Human Cases: Past

The first serologically confirmed human case of tularemia in Austria affected a woman in Mistelbach (Lower Austria) in 1934 [79]. An accumulation of human infections was observed from late November 1936 to early January 1937. Of approximately 200 affected persons in the districts of Mistelbach and Gänserndorf (Lower Austria, region north and north east of Vienna), serological confirmation was successfully performed in 94 cases. Transmission occurred during the skinning and disemboweling of hares [79]. A second cluster of human diseases was described between November 1945 and April 1946 in the same district. Human cases were also reported alongside the rivers Danube, March, and Thaya, comprising the northeastern regions of Lower Austria down near Vienna, which has remained the “natural focus” of tularemia in Austria. One year later, in 1947, sporadic human cases south of the Danube were documented. A major human tularemia epidemic occurred in 1959/1960 in sugar factories in Lower Austria, south of the Danube and close to the province of Burgenland. Following the inhalation of contaminated aerosols formed by dead, infected rodents entering the beet-washing process, a total of 577 human cases were recorded [80]. Further outbreaks of infections were observed after a strong proliferation of field mice (*Microtus arvalis*) promoted by favorable weather conditions like mild winter and hot summer temperatures [81]. Another major epidemic occurred in the late 1960s, with 170 confirmed cases of human tularemia.

#### 3.1.2. Human Cases: Present

According to the Federal Ministry of Labour, Social Affairs, Health and Consumer Protection, a total of 214 human infections with tularemia were reported in Austria over the last thirty years. Subject to a wide range of variation (from 0 to 26), the annual average incidence was 7 cases per year [82]. While the number of human infections was continuously decreasing during the years 1990 to 1993, an accumulation was then observed in 1994 (26 cases), 1995 (16 cases), 1997 (16 cases), and 1998 (19 cases), with the largest proportion occurring in the endemic area of the Danube region, especially in the province of Lower Austria. The increased incidence was reported to be associated with two simultaneously occurring field hare epizootics in the same area [83,84]. Between 1999 and 2013, fewer than 10 infections per year were consistently reported. In 2014, no human cases of tularemia were notified, while in the following years, a fluctuation of the incidence rate was observed again. A significantly above-average number of cases was recorded in 2017 (13 cases) and 2019 (20 cases). The number of cases over the last three decades, broken down by individual federal provinces, are depicted in Figure 1. A geographical summative representation of tularemia cases in humans in the various Austrian provinces from 1990 to 2019 is shown in Figure 2. Interestingly, in recent years, several infections were reported from Western Austria, especially in the province of Tyrol, where only isolated cases of human tularemia had occurred before. In 2016, a small series of tick-borne, ulceroglandular tularemia in humans was registered in Tyrol [78]. Later in 2018, the first two human cases of ulceroglandular tularemia after direct contact with an infected hare were confirmed in a hunter and a butcher near Innsbruck, Tyrol (Appendix A). 

### 3.2. F. tularensis in Animals

In Austria, *F. tularensis* has been detected in hares, wild rabbits, mice, and ticks [51,53,83,84,85]. Due to their status as common vectors for tularemia, prevalence studies have been performed most frequently in field hares. 

#### 3.2.1. *F. tularensis* in Hares

It is assumed that transmission to hares may occur as a consequence to preceding mouse epizootics [86]. In 1936, a field hare at the border of former Czechoslovakia tested positive for tularemia for the first time [85]. In 2003, Winkelmayer et al. examined a total of 311 serum samples derived from clinically healthy European field hares from six Lower Austrian districts (northeastern region comprising the “natural focus”) for antibodies against *F. tularensis*. Of these serum samples, 7.1% tested positive, which is in good agreement with earlier results by Höflechner-Pöltl et al. in 2000 (positive detection in 4.5%). However, it must be noted that the latter study only included clinically ill animals and animals killed in accidents [87,88]. Between 1994 and 2005, a total of 271 hares tested positive for *F. tularensis* in Lower Austria, Burgenland, and Styria [89].

#### 3.2.2. *F. tularensis* in Foxes

In 2007, *F. tularensis* was detected for the first time in the mandibular lymph nodes of red foxes (*Vulpes vulpes*) from northern Burgenland that were collected by hunters as part of the rabies control program [90]. Between 2007 and 2008, Hofer and colleagues investigated mandibular lymph nodes of 903 red foxes from the provinces of Lower Austria, Burgenland, Upper Austria, and Styria. Infection with *F. tularensis* was detected in only 10 animals, and the distribution of the positive findings corresponded to the known endemic areas in Eastern Austria [91].

#### 3.2.3. *F. tularensis* in Ticks

Experimental animal transmission tests for *F. tularensis* were performed with ticks from Austria in 1998. In this study, 2.8% of *Dermacentor reticulatus* ticks from Lower Austria transmitted tularemia to mice [51]. Furthermore, between 1991 and 1997, up to 1.3% of *Dermacentor reticulatus* ticks collected in regions of northeastern Austria were infected with *F. tularensis* [53]. In a study in Baden-Württemberg (Germany), which geographically borders Western Austria, a 16S rRNA-gene screening by PCR of 95 pools of *Ixodes ricinus* ticks showed a positive signal in 8.4% [92]. Transmission of tularemia via ticks to humans was first reported in Western Austria [78].

#### 3.2.4. *F. tularensis* in Domestic Animals

There is only limited data in terms of tularemia in domestic animals in Austria. An examination of 80 clinically healthy dogs, used for hunting in the endemic area of Lower Austria, showed a seroprevalence for *F. tularensis* in 6.25% [93]. According to our literature search, no human infections transmitted by domestic animals have been reported in Austria so far.

## 4. Discussion

Tularemia is a rare infection in humans in Austria. Low numbers have been reported throughout the last 85 years, though sporadic epidemic outbreaks have occurred, all of them originating from the North and Northeast of Austria. From the districts north of the Danube in Lower Austria, *F. tularensis* spread further south to Burgenland and Styria; thus, for many years, this region of Austria was regarded the “natural or endemic focus” of tularemia. Since then, hares have been considered to play an important role in transmission of tularemia in Austria, and humans have been assumed to be infected when skinning and handling the carcasses [79]. Since 1991, an official registry of human infections is accessible in Austria. While some cases of active natural endemic areas in Eastern Austria are continuously being reported, until recently, single (likely imported) human cases were diagnosed in Western Austria (Vorarlberg, Tyrol, and Salzburg), which had been regarded as free of tularemia in the past. In 2016, a series of human tularemia cases were reported from Tyrol, and all these infections were transmitted via tick bites, a way of transmission that had not been reported before in Austria [78]. Thus, the question arises whether ticks were underestimated vectors in the endemic regions around Danube, March, and Thaya and had moved from east to west. Alternatively, a new endemic area in Western Austria could have evolved due to the migration of infected ticks from the north (Germany) and west (Switzerland), where they had previously been described as the main vectors of tularemia [94,95]. Interestingly, *F. tularensis* subsp. *holarctica* biovar I was identified in one Tyrolean patient for the first time in Austria [78]. The distribution of biovar I and II in Western Austria is in line with the prevalent data on the coexistence of both biovars in Germany and Switzerland, and demonstrates an epidemiological spread of different clades of *F. tularensis* subsp. *holarctica* in these countries [96,97,98].

In 2016, based on genetic and geographical analyses, Dwibedi et al. demonstrated that *F. tularensis* had spread from east to west within Europe [18]. Environmental and climatic factors seem to play a key role in the incidence of tularemia. In 2009, Deutz and colleagues investigated the impact of climate and weather on the geographical distribution of tularemia in Austria, and demonstrated a positive correlation to mild winters, cool springs, and a high precipitation amount in summer. They calculated a 3.5-fold increase of the tularemia endemic area in Austria, with an expansion from east to west by 2035 due to global warming [89]. An important contributing factor is the effect of climate change on population density, as well as the susceptibility of field hares (especially young animals) to pathogens, as these still play a key role in Austria as natural hosts for *F. tularensis*. In turn, weakening of young animals makes them easier prey for natural enemies, such as the red fox, thus favoring transmission between animals of different species with a greater range of movement and concomitant spread of tularemia [89]. The reported human infections caused by the handling of a hunted hare indicate that this species has now become an established reservoir for *F. tularensis* in Western Austria. From April 2018 to May 2020, the pathogen was detected in 38 European brown hares, sent to the Institute for Veterinary Disease Control of the Austrian Agency for Health and Food Safety (AGES) in Mödling by official veterinarians, indicating epizootics in hares in the federal states Salzburg, Tyrol, and Vorarlberg. Thirty cases of tularemia from hares were found in Salzburg, seven in Vorarlberg, and one in Tyrol (unpublished data). These data suggest that the expansion of the *F. tularensis* habitat, which was estimated to not be reached before 2035, has now already been attained [89]. Prevalence data of *F. tularensis* in Austrian ticks, as well as in mosquitos or horse flies, are very scarce or completely absent. As these are known potential vectors of tularemia to humans, further investigation is required. In order to improve the epidemiologic knowledge of tularemia in wild animals and the associated risks to humans, long-term systemic monitoring of known natural reservoir animals will be essential in the future.

## 5. Conclusions

Previous studies on the prevalence of *F. tularensis* in field hares, ticks, and to a lesser extent, red foxes have been limited to the known endemic areas in Eastern Austria, and therefore allow only a very limited estimation of the actual natural spread of tularemia in Austria. Given the knowledge that epizootics precede epidemics of tularemia in humans, long-term systemic surveillance of natural foci will be essential to monitor the distribution and prevalence of *F. tularensis*, in order to anticipate possible risks of outbreaks and to take appropriate preventative measures [53]. This requires the broad screening of potential reservoir animals for *F. tularensis*, in order to get a real image of distribution of tularemia in Austria, which can only be successful through close cooperation of humans, veterinary health professionals, and authorities. It is furthermore important to raise awareness of tularemia among the general public, particularly for risk groups such as hunters, foresters, butchers, and taxidermists. Finally, doctors must be trained to be aware of this infection with its various forms of presentation [99].

## Figures and Tables

**Figure 1 microorganisms-08-01597-f001:**
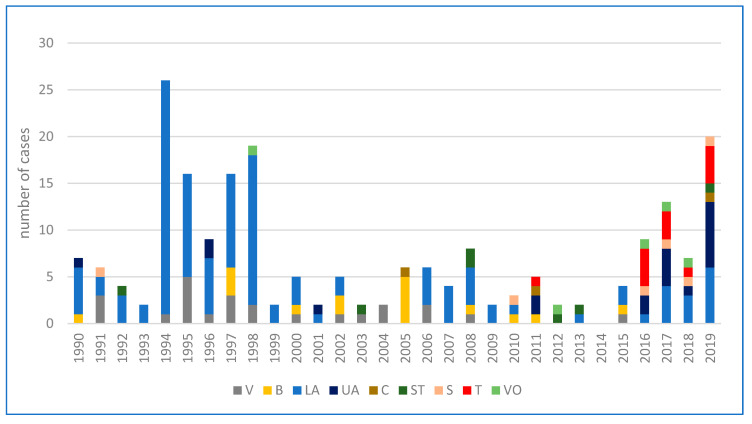
Cumulative cases of human tularemia in Austria divided by provinces from 1990–2019. Abbreviations: V = Vienna, B = Burgenland, LA = Lower Austria, UA = Upper Austria, C = Carinthia, ST = Styria, S = Salzburg, T = Tyrol, VO = Vorarlberg [82].

**Figure 2 microorganisms-08-01597-f002:**
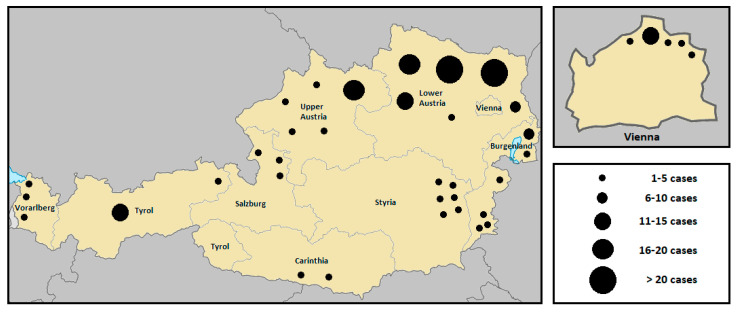
Summative geographic distribution of human tularemia cases in the various Austrian provinces from 1990 to 2019. The size of the dots corresponds to the number of humans infected with *F. tularensis* during a five year period [82].

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
