# Peer review of "Tularemia Goes West: Epidemiology of an Emerging Infection in Austria"

_microorganisms, 2020, doi:10.3390/microorganisms8101597_

Round 1
Reviewer 1 Report
This review describes epidemiology of tularemia in Austria. This is new and has not been reviewed in detail recently, so I think it is well worth to publish and think the review is of general interest to the Francisella community. However, I have several remarks that needs to be addressed
- Introduction, line 35, add a reference on the division of the four subspecies, different classifications are reported in the literature, the debate concerns whether subsp novicida should be an own subspecies or not.
- Introduction, 39-40, Here the division in major genetic clades for type B should be described, where B.16 corresponds to biovar japonica and B.12 corresponds to biovar II, B.12 dominates in Austria, but also a B.11 (a subclade of major clade B.6) occurs sporadically see e.g. Pilo P. 2018. Phylogenetic Lineages of Francisella tularensis in Animals. Front Cell Infect Microbiol 8:1–11 and Wittwer et al. 2018. Population Genomics of Francisella tularensis subsp. holarctica and its Implication on the Eco-Epidemiology of Tularemia in Switzerland. Front Cell Infect Microbiol 8:1–16.
- Introduction, line 44, this could be misunderstood as if US may be responsible for the release of a laboratory strain in Europe, could be sensitive, needs to be rephrased. The SchuS4 strain was exchanged with former Soviet Union in 1956.
- Introduction, line 53-62. The ecology of tularemia is extremely complex and I think the description here is an oversimplification, especially the statement that the European hare (Lepus europaeus), is the reservoir species in Austria. More reasoning on the complexity and alternatives should be discussed. I recommend that some of the views described in Telford and Goethert. 2010. Toward an understanding of the perpetuation of the agent of tularemia. Front Microbiol 1:150 and Telford and Goethert. 2020. Ecology of francisella tularensis. Annu Rev Entomol 65:351–372 are mentioned. Traditionally, Austria belong to the ticks-hares ecological system (HJ Jusatz) where also the field mouse play a central role for the transmission to hares. A fact, that is illustrated by the 1959/1960 outbreak which is described in section 3.1.1. Further it is mentioned the recent occurrence of tick-borne tularemia in Austria, corroborating the traditional view of the ticks-hares ecological system in Austria. In other parts of Europe, (France and Sweden) the European brown hare present acute form of the disease (Decors et al.2011, Hestvik et al, 2017), thus indicating that the ecological system are more complex and probably consist of several layers of multi-host interactions. Further, it should be mention that the different disease presentations could possibly be due to different genotypes of type B strains (for a discussion see Pilo et al 2018).
- Discussion, line 200,replace hares as main reservoirs with something like “hares plays the main role as a source of infection” Throughout the manuscript I would be careful to state that European hares are the main reservoir of disease, this would imply that if the hares is taken out of the system the spread of tularemia would stop. As we know this is probably not the case. One example of newly formed ecosystem for tularemia is Spain, where landscaping created a “new” epidemiology context which is special in almost absence of vector-borne transmission, instead the majority of human cases are by aerosol or direct contact with voles, i.e. a system that disseminate tularemia locally in the absence of both hares and ticks (Rodríguez-Pastor et al., 2017).
- Discussion line 240, there is not such term as approved vectors for tularemia, compare with the reports by Petersen JM, Mead PS, Schriefer ME. 2009. Francisella tularensis : an arthropod-borne pathogen. Vet Res 40:07. Pilo P. 2018. Arthropod Infection Models for Francisella tularensis. Another insightful source for discussion on the role ticks as vectors is Zellner B, Huntley JF. 2019. Ticks and tularemia: Do we know what we don’t know? Front Cell Infect Microbiol 9.
- Use the term potential reservoir animals or something less definitive. .
Reviewer 2 Report
The manuscript entitled “Tularemia goes West – epidemiology of an emerging infection in Austria” and submitted as a review provides information on the prevalence and spread of tularemia in Austria, along with some background information about this zoonotic disease. However, the unorganized information related to tularemia in Austria makes it difficult to understand and the message that the authors are trying to convey in each paragraph is unclear. Moreover, the figures are either incomplete, redundant, and/or non-informative. The incorrect usage of the English language and unclear or inaccurate sentences throughout the manuscripts made reviewing this article very difficult. The amount of information presented for the spread of F. tularensis subsp. holarctica in Austria is limited, and therefore, qualifies as a communication rather than a review article, if major revision makes this manuscript worthy of publication. Some of the revisions needed include the following.
1.) Names of species and subspecies should be italicized throughout the manuscript.
2.) The usage of the English language, punctuation, and misspellings/typos (e.g., Oktober) need to be corrected throughout the manuscript, including the abstract.
3.) The sentence on lines 29-31 is not accurate, since not all subspecies of F. tularensis are highly virulent.
4.) The authors need to expound on glycerol fermentation by biovar japonica compared to other type B biovars or strains (lines 36-40).
5.) It is unclear who the authors are referring to in the pronoun “they” on line 45, and this overall sentence seems to be irrelevant with only one F. tularensis subsp. tularensis strain (FSC198) sequenced and only one reference cited. In addition, this sentence contradicts the following sentence, which states that “tularemia in Europe has exclusively been caused by F. tularensis subsp. holarctica”.
6.) On lines 66-67, the reference cited was written in 1993, while the authors state “so far, 51 cases of cat-related tularemia have been reported”. More current references and the number of cat-related tularemia cases to date is needed.
7.) The two sentences on lines 75-78 contradict each other. Please correct or clarify.
8.) The abbreviation “EU/EAA” should be written out.
9.) The usage of LVS as a vaccine should be discussed, especially if this manuscript is submitted as a review.
10.) The statement on lines 98-100 needs to be supported by references.
11.) The usage of “notifiable” that is often used in the manuscript should be explained when first used.
12.) Sentence on lines 142-144 needs rewording for clarification.
13.) Lines 146-148 discuss methods and should be moved to the methods section.
14.) Figure 1b is redundant to Figure 1a, and doesn’t offer any additional information.
15.) A map of the states with the cases of tularemia described in Figure 1a would be valuable.
16.) Figure 2a-c doesn’t provide any useful information and takes up a lot of space.
17.) The section on “Human cases - present” (lines 146-164) is poorly organized and unclear. In addition, hares are not rodents (lines 158-161), so the associated sentence needs to be corrected.
18.) Sentence on lines 186-187 is unclear and needs clarification, along with more details on the cases of tularemia being described.
19.) The phrase on lines 216-217 is unclear and needs clarification.
20.) The reference for the sentences on lines 220-225 needs to be cited, and this 2009 reference is too old for the following statement on lines 225-227.
21.) Sentence on lines 230-232 is unclear and needs clarification.
Author Response
Please see the attachment.
Reviewer's comments and point-by-point responses
Thank you for reading and reviewing our manuscript. We checked and corrected the manuscript carefully for mistakes and inserted the amendments according to your comments. Please find listed below our point-by-point-responses.
Point 1: Names of species and subspecies should be italicized troughout the manuscript.
Response 1: This was corrected throughout the manuscript.
Point 2: The usage of the English language, punctuation and misspellings/typos (e.g., Oktober) need tob e corrected throughout the manuscript, including the abstract.
Response 2: Typing mistakes were corrected and the manuscript was proofread by a native speaker.
Point 3: The sentence on lines 29-31 is not accurate, since not all subspecies of F. tularensis are highly virulent.
Response 3: Thank you for pointing this out. We removed the term "highly-virulent" as we refer to the virulence of the subspecies tularensis and holarctica in a separate section of the introduction (lines 28, 37 and 50).
Point 4: The authors need to expound on glycerol fermentation by biovar japonica compared to the other type B biovars and strains (lines 36-40).
Response 4: This was endorsed according to the literature (lines 40-42).
Point 5: It is unclear who the authors are reffering to in the pronoun “they“ on line 45, and this overall sentence seems to be irrelevant with only one F. tularensis subsp. tularensis strain (FSC198) sequenced and only one reference cited. In addition, this sentence contradicts the following sentence, which states that “tularemia in Europe has exclusively been caused by F. tularensis subsp. holarctica“.
Response 5: We herein report the unique detection of F. tularensis subsp. tularensis in Austria und its interpretation. The sentences were clarified in lines 51-56.
Point 6: On lines 66-67, the reference cited was written in 1993, while the authors state “so far, 51 cases of cat-related tularemia have been reported“. More current references and the number of cat-related tularemia cases to date is needed.
Response 7: We have added several more recent references and adapted the text (lines 103-107).
Point 7: The two sentences on lines 75-78 contradict each other. Please correct or clarify.
Response 7: The introduction section was reorganised and the sentences were corrected (lines 28-139).
Point 8: The abbrevation “EU/EAA“ should be written out.
Response 8: The abbreviations for European Union (EU) and the European Economic Area (EAA) were explained (lines 138-139).
Point 9: The usage of LVS as a vaccine should be discussed, especially if this manuscript is submitted as a review.
Response 9: This was elaborated in lines 126-132.
Point 10: The statement on lines 98-100 needs to be supported by references.
Response 10: The according reference was inserted (lines 134-137).
Point 11: The usage of “notifiable“ that is often used in the manuscript should be explained when first used.
Response 11: The explanation was inserted in lines 133-134.
Point 12: Sentences on lines 142-144 needs rewording for clarification.
Response 12: The wording was specified in lines 177-179.
Point 13: Lines 146-148 discuss methods and should be moved to the methods section.
Response 13: This was corrected in lines 141-147.
Point 14: Figure 1b is redundant to Figure 1a and doesn’t offer any additional information.
Response 14: Figure 1b was removed.
Point 15: A map of the states with the cases of tularemia described in Figure 1a would be valuable.
Response 15: A Map of the Austrian states was depicted with dots indicating the number of cases (now Figure 1b).
Point 16: Figure 2a-c doesn’t provide any useful information and takes up a lot of space.
Response 16: Figure 2b was removed and Figures 2a and c moved to supplementary materials.
Point 17: The section on “Human cases – present“ (lines 146-164) is poorly organized and unclear. In Addition, hares are not rodents (lines 158-161), so the associated sentence needs to be corrected.
Response 17: The section was reorganized and the correction of hares as lagomorphs was made (lines 183-201).
Point 18: Sentence on lines 186-187 is unclear and needs clarification, along with more detailes on the cases of tularemia being described.
Response 18: This was clarified in lines 232-235.
Point 19: The phrase on lines 216-217 is unclear and needs clarification.
Response 19: The wording was specified in lines 264-265.
Point 20: The reference for the sentences on lines 220-225 needs to be cited and this 2009 reference is too old for the following statement on lines 225-227.
Response 21: The sentence was clarified cited in lines 267-271.
Point 21: Sentence on lines 230-232 is unclear and needs clarification.
Response 21: The phrasing was clarified (lines 276-278).
Reviewer 3 Report
General comments
The manuscript of Seiwald et al. is very well written. The review well describes the epidemiology of Francisella to be an emerging infection also in Austria. It describes that Francisella now has been spread all-over Austria, a strain of biovar I has been identified for the first time and ticks as a vector for tularemia.
I have only some minor comments:
- The abstract has 10 words more than "prescribed"
- lines 33-50: please italicise the text where needed
- line 52: I suggest the introduced new paragraph is not needed
- line 64: you may include the reference "Zellner and Huntley, 2019" / ticks
- line 67: you may include the reference "Kwit et al., 2019" / dogs
- line 71: "scarcing" ?
- line 99: "United Kongdom"
- Fig 1a: colour indicating "B" and "T" are indistinguishable
- Fig. 2: b and c seems to be inverted; picture of the "enlarged axillar lymph node" is not easy to interpret for medical ordinary person without further labelling or markers
- line 67: you may include the reference "Appelt et al., 2019" / biovars in Germany
Author Response
Thank you for reading and reviewing our manuscript and the valuable comments. Please find our point-by-point-responses listed below.
Point 1: The abstract has 10 words more than prescribed.
Response 1: Thank you for noticing. We have shortened the abstract to the prescribed maximum word count (lines 13-25, 189 words).
Point 2: lines 33-50. Please italicise the text were needed.
Response 2: We italicized the names of species and subspecies throughout the manuscript.
Point 3: line 52: I suggest the introduced new paragraph is not needed.
Response 3: The paragraph was removed. The introduction was reorganised (lines 28-139).
Point 4: line 64: You may include the reference "Zellner and Huntley, 2019"
Response 4: Thank you for pointing out this paper to us. We have inserted the corresponding reference (lines 78-80).
Point 5: line 67: You may include the reference "Kwit et. al., 2019" / dogs
Response 5: The corresponding reference was inserted (lines 103-106). We also supplemented an additional reference with recent data on cat-associated tularemia in Nebraska, USA (lines 106-107).
Point 6: "Scarcing"
Response 6: Thank you for noticing that mistake. One way of transmission to humans is via direct contact to infected animals, i.e. skinning and processing. “Scarcing“ was replaced to “animal processing“ because it summarizes several actions related to handling the dead animal (line 65).
Point 7: „United Kongdom“
Response 7: We corrected the spelling mistake (line 135) and checked the manuscript carefully for further typing or spelling errors as well as grammatical deficiencies.
Point 8: Fig. 1a: colour indicating “B“ and “T“ are indistinguishable.
Response 8: Thank you for pointing this out. We changed the colour indicating “T“ to red. Now all used colors stand out clearly from each other (lines 202-203).
Point 9: Fig. 2: b and c seems to be inverted; picture of the enlarged axillar lymph node is not easy to interpret for medical ordinary person without further labelling or markers.
Response 9: We have corrected the names of the images. According to your considerations and the comments of reviewer 1 we have removed the image of the "enlarged lymph node" and figure 2 is now listed as supplementary material.
Point 10: Line 67: You may include the reference “Appelt et al., 2019 / biovars in Germany.
Response 10: Thank you for this recommendation, we added the reference also to lines 260-263.
Round 2
Reviewer 1 Report
I am satisfied with the response and changes. The author have done a fine job addressing the issues I raised. However, I recommend going through the reference list, there are some missing text and inconsistencies in style and font.
Reviewer 2 Report
Reviewer Comments and Suggestions for Authors
The revised version 2 of the manuscript entitled “Tularemia goes West – epidemiology of an emerging infection in Austria” provides information on the prevalence and spread of tularemia in Austria, along with some background information about this zoonotic disease. The revised manuscript has been considerably improved. However, there are still numerous sentences with the incorrect usage of the English language, lack clarity, and/or are inaccurate. In conclusion, this manuscript still needs additional revision before it is worthy of publishing in Microorganisms.
Areas in the revised version 2 of this manuscript that need correction include the following.
1.) The statement on lines 34 and 35 is not accurate and needs to be corrected.
2.) The word “West” in the title and the word “Type” on line 54 should not be capitalized.
3.) The sentence on lines 57 and 58 is out of place and doesn’t appear to belong to the previous nor the following paragraph, based on the formatting/indentations.
4.) The long sentence on lines 63 through 67 is unclear and the wording has punctuation and grammar issues.
5.) The long sentence on lines 71 through 74 needs to be corrected for grammar and shortened for clarity.
6.) The authors should describe what differences in pathogenicity were observed between the specified clades, which are briefly mentioned in the sentence on lines 74-77.
7.) The wording in the two sentences on lines 85-88 needs to be corrected.
8.) The phrase “acquire infection” should be changed to “be infected” on line 93.
9.) The word “Nebraska” on line 107 should be changed to “USA”, since the described feline cases of tularemia in this reference were also from other states in the midwest and east.
10.) The phrase “serological-, PCR-” on line 122 should be revised to “serology, PCR” and the word “culture” on line 123 should be changed to “from culturing” for clarity.
11.) The word “lacking” on line 124 should be changed to “lack of” for proper English grammar.
12.) The statement on lines 128-131 is not accurate, since the LVS vaccine does not protect against type A strains. Therefore, this sentence needs to be corrected.
13.) The sentence on lines 149-150 is unclear and needs rewording for clarity.
14.) Please delete the phrase “A first” on line 166 and the word “newly” on line 174.
15.) The phrase “Such a” on lines 179 should be changed to “Another” for clarity.
16.) The sentence on lines 195-196 (“A geographical summative representation of the human cases, summarized in 5 years periods each is shown in Figure 1b.”) should be changed to “A geographical summative representation of tularemia cases in humans in the various Austrian provinces from 1990-2019 is shown in Figure 1b.”.
17.) The sentence on lines 198-199 needs rewording.
18.) Please delete the word “hunted” on line 200.
19.) The phrase “summarized in” on lines 210 in the figure legend should be changed to “during”.
20.) The phrase “a as” on line 216 should be “as a”.
21.) Delete the word “was” on both lines 217 and 224, and delete the word “were” on line 221.
22.) Change the phrase “of regularly shot,” on line 219 to “derived from”.
23.) The percentage of “7.1%” needs to be written on line 221, since it is at the beginning of the sentence.
24.) Delete the sentence “With the European hares as prey, F. tularensis was also investigated in predators” on lines 224 and 225, since other vectors can transmit this pathogen.
25.) It is unclear if the phrase “ten animals” on line 230 is still referring to foxes. Please clarify and if this is the case, perhaps divide this large section/paragraph into three paragraphs to describe tularemia in hares, foxes, and ticks.
26.) Delete the word “Moreover” on line 231 and state if “animal transmission tests” refers to experimentally controlled studies or natural environmental cases.
27.) Insert the word “by” in front of the word “PCR” on line 236.
28.) The sentence on lines 239-241 needs rewording due to poor grammar.
29.) Delete the word “with” on line 250 and the word “Only” on line 253.
30.) Change the word “notified” on line 254 to “reported” for clarity.
31.) The long sentence on lines 260-263 needs to be shortened and is inaccurate. Perhaps change the word “subspecies” on lines 262 to “clades” to correct this statement.
32.) The three sentences on lines 265-271 are unclear. Please reword for clarity and define what is meant by “associated precipitation amount”.
33.) Delete three commas in the sentence on lines 276-278.
34.) Change the phrase “be reached not” to “not be reached” on line 283, and change “in now already attained” in this same sentence to “has now already been attained”.
35.) Perhaps move the sentence on lines 287-289 to follow the previous sentence, rather than starting a new paragraph that only contains this single sentence.
36.) Change the phrase “human and” to “humans,” on line 300.
37.) Delete the word “regarding” on line 302.
Round 3
Reviewer 2 Report
The third revision of the manuscript entitled “Tularemia goes West – epidemiology of an emerging infection in Austria” provides information on the prevalence and spread of tularemia in Austria, along with some background information about this zoonotic disease. The third revision of this manuscript has been substantially improved relative to the first and second version and now only has minor issues that can be easily corrected. After these minor corrections, this manuscript is worthy of publishing in Microorganisms.
The minor issues that need correction in this 3rd manuscript revision are listed below.
1.) On line 147, add the word “the” in front of “USA”.
2.) Add the “stand-alone” sentence on lines 149-151 to the following paragraph.
3.) Add the word “the” in front of the word “lack” on line 164.
4.) Indent new paragraph on line 173.
5.) Add the sentence on lines 209-212 to the previous paragraph.
6.) Delete the word “respectively” on line 251.
7.) Reword the sentence on lines 272-273 in the figure legend to the following: “The size of the dots corresponds to the number of humans infected with F. tularensis during a five year period.”
8.) Reword the first part of the sentence on lines 285-286 as follows: “Of these serum samples, 7.1% tested positive,”.
9.) Add a blank line after the sentence on line 297 and remove the word “to” on line 303.
10.) The word “research” on line 327 should be changed to “search”.
11.) Remove the word “Only” on line 340 and change the phrase “a small case series of human tularemia was reported” on line 340 to “a series of human tularemia cases were reported”.
12.) Delete the word “also” on line 342.
13.) Add the word “the” in front of the word “north” on line 344.
14.) One line 363, remove the words “also” and “as”, and add the word “an” in the word “established”.
15.) Change the phrase on lines 414-415 from “thereby anticipating possible risks of outbreaks and taking preventive measures” to the following: “in order to anticipate possible risks of outbreaks and to take appropriate preventative measures”.
Author Response
Thank you for reading and reviewing our resubmitted manuscript and the further valuable comments to improve our manuscript. Please find our point-by-point-responses listed below (As the current change requests referred exclusively to grammar, style and expression, no further explanation was given).
Point 1: On line 147, add the word “the” in front of “USA”.
Response 1: Done (line 124).
Point 2: Add the “stand-alone” sentence on lines 149-151 to the following paragraph.
Response 2: Done (lines 125-128).
Point 3: Add the word “the” in front of the word “lack” on line 164.
Response 3: Done (line 141).
Point 4: Indent new paragraph on line 173.
Response 4: A new paragraph was added to line 152.
Point 5: Add the sentence on lines 209-212 to the previous paragraph.
Response 5: The sentence in lines 179-182 was added to the previous paragraph.
Point 6: Delete the word “respectively” on line 251.
Response 6: Done (line 215).
Point 7: Reword the sentence on lines 272-273 in the figure legend to the following: “The size of the dots corresponds to the number of humans infected with F. tularensis during a five year period.”
Response 7: The sentence was reworded as suggested (lines 233-235).
Point 8: Reword the first part of the sentence on lines 285-286 as follows: “Of these serum samples, 7.1% tested positive,”.
Response 8: Done (lines 247-248).
Point 9: Add a blank line after the sentence on line 297 and remove the word “to” on line 303.
Response 9: Revised as requested (lines 262 and 268).
Point 10: The word “research” on line 327 should be changed to “search”.
Response 10: Done (line 278).
Point 11: Remove the word “Only” on line 340 and change the phrase “a small case series of human tularemia was reported” on line 340 to “a series of human tularemia cases were reported”.
Reponse 11: Revised as requested (lines 291-292).
Point 12: Delete the word “also” on line 342.
Response 12: Done (line 294).
Point 13: Add the word “the” in front of the word “north” on line 344.
Response 13: Done (line 296).
Point 14: One line 363, remove the words “also” and “as”, and add the word “an” in the word “established”.
Response 14: The sentence was changed as requested (line 323).
Point 15: Change the phrase on lines 414-415 from “thereby anticipating possible risks of outbreaks and taking preventive measures” to the following: “in order to anticipate possible risks of outbreaks and to take appropriate preventative measures”.
Response 15: The sentence was changedd as requested (lines 342-344).